# Profiles of lipid, protein and microRNA expression in exosomes derived from intestinal epithelial cells after ischemia-reperfusion injury in a cellular hypoxia model

Atsushi Senda[1], Mitsuaki Kojima[1,2]*, Arisa Watanabe[3], Tetsuyuki Kobayashi[3], Koji Morishita[1], Junichi Aiboshi[4], Yasuhiro Otomo[1]

1 Department of Acute Critical Care and Disaster Medicine, Graduate School of Medical and Dental Sciences, Tokyo Medical and Dental University, Bunkyo-ku, Tokyo, Japan, 2 Emergency and Critical Care Center, Tokyo Women's Medical University Adachi Medical Center, Adachi-ku, Tokyo, Japan, 3 Department of Biological Sciences, Graduate School of Humanities and Sciences, Ochanomizu University, Bunkyo-ku, Tokyo, Japan, 4 Department of Emergency and Critical Care Medicine, Tokyo Women's Medical University Yachiyo Medical Center, Yachiyo, Chiba, Japan

* kojima.mitsuaki.twmu@ac.jp

**Data Availability Statement:** All relevant data are within the paper and its Supporting Information files.

## Abstract

Intestinal ischemia-reperfusion injury leads to proinflammatory responses via gut-derived mediators, and accumulating evidence suggests that exosomes secreted by intestinal epithelial cells are involved in the development of systemic inflammation. Studies have reported changes in protein, lipid, and microRNA (miRNA) expression; however, considering the different experimental conditions, information on the relationships among these biomolecules remains insufficient. The aim of this study was to elucidate the multiple changes that simultaneously occur in exosomes after ischemic stimulation. Here, differentiated human intestinal Caco-2 cells were exposed to 95% air (normoxia group) or 5% $O_2$ (hypoxia group) for 6 h. Cells in each group were subsequently incubated for 24 h in an atmosphere of 5% $CO_2$ plus 95% air. The conditioned medium of each group was collected for isolating intestinal epithelial cell-derived exosomes. Together with proteome analyses, lipid analyses, and miRNA quantification, biological functional assays were performed using monocytic NF-κB reporter cells. Lipid metabolism-related protein expression was upregulated, miRNA levels were slightly altered, and unsaturated fatty acid-containing lysophosphatidylcholine concentration increased after hypoxia and reoxygenation injury; this suggested that the changes in exosomal components associated with ischemia-reperfusion injury activates inflammation, including the NF-κB pathway. This study elucidated the multiple changes that co-occur in exosomes after ischemic stimulation and partially clarified the mechanism underlying exosome-mediated inflammation after intestinal ischemic recanalization.

## Introduction

Intestinal ischemia is a life-threatening condition associated with a broad range of clinical conditions, including trauma, mesenteric ischemia, septic shock, and bowel inflammatory

**Funding:** MK, Scientific Research from the Japan Society for the Promotion of Science (18K16509) AS, Scientific Research from the Japan Society for the Promotion of Science (21K16585).

**Competing interests:** The authors have declared that no competing interests exist.

diseases. It is known to cause multiple organ failure after intestinal ischemia-reperfusion injury [1], with dysregulated inflammatory response considered the main cause [2, 3]. This response can be prevented by the ligation of the mesenteric lymph (ML) duct [4]; therefore, ML is considered to have a pathologically crucial role [4–7]. Various ML components have been identified as potential causative agents of this response but have not yet been confirmed. In recent years, an increasing number of studies have suggested that exosomes are the primary causative agent.

Exosomes are nanosized (30–150 nm) extracellular vesicles [8] that carry microRNAs (miRNAs), proteins, and lipids [9]. They have been demonstrated to upregulate monocyte nuclear factor (NF)-κB expression and increase macrophage intra-cellular TNF-α levels, which subsequently cause lung injury [10, 11]. A few studies have identified the exosome cargo responsible for this reaction. One *in vivo* study identified that polyunsaturated fatty acids containing lysophosphatidylcholine (LPC) upregulate and induce NF-κB expression [12]. Another *in vivo* study showed changes in the protein composition of exosomes following ischemic stimulation [13]. These results suggest that ischemic stimuli induce alterations in various exosome components that contribute to the subsequent inflammatory response. Therefore, for integrated understanding of the pathogenesis of systemic inflammation, it is necessary to simultaneously capture the changes in multiple components. To achieve this purpose, we intended to establish an *in vitro* intestinal ischemia-reperfusion injury model as it is currently difficult to harvest a sufficient amount of exosomes from ischemia-reperfusion small-animal models.

Given that ML exosomes mainly originate from intestinal epithelial cells (IECs) [11], we developed an *in vitro* experimental system that mimics ischemia-reperfusion injury. Corresponding to the ischemic state, cultured IECs were first placed under hypoxic conditions. Subsequently, they were returned to an oxygenated environment corresponding to the reperfusion state. Accordingly, this hypoxia reoxygenation model has been used to simulate ischemia-reperfusion injury in an *in vitro* setting in previous studies [14, 15]. This facilitated integrated understanding of the changes that occur in the exosome following ischemic stimulation.

## Materials and methods

### Study design

Differentiated human intestinal Caco-2 (ATCC HTB37) cells were exposed to either 5% $CO_2$ and 95% air (normoxia group) or 5% $O_2$, 5% $CO_2$, and 90% $N_2$ (hypoxia group) for 6 h. Then, the culture medium was changed. To place the cells in an environment analogous to the reperfusion period during hemorrhagic shock, each group was subsequently incubated in a normoxic environment (5% $CO_2$ and 95% air) for 24 h. The conditioned medium of each group was collected for isolating IEC-derived exosomes. Subsequently, biological function assays were performed using monocytic NF-κB reporter cells, proteome analysis was conducted using high-performance liquid chromatography–mass spectrometry (HPLC–MS), and miRNA quantification was conducted using qPCR (S1 Fig).

### Cell culture

Caco-2 is a carcinoma epithelial cell line of colonic origin with small intestinal enterocyte-like features [16]. Cells were cultured in Dulbecco's modified Eagle medium (Nakaraitesk, Kyoto, Japan) supplemented with 2 mM L-glutamine (Gibco, Thermo Fisher Scientific, Waltham, MA, USA), 100 U/mL penicillin, 100 μg/mL streptomycin, and 10% (v/v) fetal bovine serum (FBS). THP-1 is a monocytic cell line engineered for monitoring NF-κB activation by determining the activity of secreted embryonic alkaline phosphatase. The growth medium consisted of RPMI-1640 supplemented with 2 mM L-glutamine, 25 mM HEPES, 10% (v/v) heat-inactivated FBS, 100 μg/mL normocin, and 100 U/mL penicillin-100 μg/mL streptomycin. Both cell

types were transferred to FBS-free medium 72 h before the experiments to avoid contamination [17].

## Isolation of exosomes

To avoid contamination, exosomes were isolated from serum-free conditioned medium (Opti-MEM) of IECs according to the method described previously, with some modifications [18]. First, the conditioned medium was centrifuged at $3,000 \times g$ for 15 min; then, the supernatant was passed through a 0.2-μm filter. Next, the medium was centrifuged using the Amicon Ultra-15 (Sigma-Aldrich, St. Louis, MO, USA) Centrifugal Filter Unit with a 100 kDa cutoff at $4,000 \times g$ for 20 min at 25°C for concentration [18]. Through these concentration methods, 80 mL of each media was concentrated into 5 mL. Subsequently, 1 mL of the ExoQuick-TC Exosome Precipitation Solution (System Biosciences, Palo Alto, CA, USA) was added to each sample, and the mixture was incubated overnight for exosomal extraction following the manufacturer's instructions.

## Immunoblotting of exosomal surface proteins

The isolated exosomes were verified via immunoblotting with the Exo-Check Antibody Array (System Biosciences), containing pre-printed spots of eight antibodies against known exosome markers (FLOT1, ICAM, ALIX, CD81, CD63, EpCAM, TSG101, and ANXA5). Samples containing 50 μg of exosome proteins were prepared. The exosome proteins were quantified using the Qubit fluorometer. The samples were subsequently diluted and reacted with the labeling reagent and the blocking buffer according to the manufacturer's instructions. The labeled exosome lysate/blocking buffer mixture was incubated with the membrane containing pre-printed spots of the eight antibodies at 5°C overnight on a rocker. After removing the lysate/blocking buffer mixture, washing was performed using a washing buffer. Then, the membrane was incubated for 30 min with detection buffer. Finally, a picture of the membrane was captured using an EOS 80D camera (Canon, Tokyo, Japan). All reagents mentioned above were included in the Exo-Check Antibody Array (System Biosciences) and the experimental conditions were maintained following the manufacturer's instructions. The blot/gel image data are original and have not been cropped or adjusted.

## Transmission electron microscopy (TEM)

Exosomal specimens for TEM were prepared by the personnel of the Instrumental Analysis Division at the Research Center of Tokyo Medical and Dental University. The exosomes were immersed in Karnovsky's fixative (pH 7.42, 2.5% glutaraldehyde, and 2% paraformaldehyde in 0.15 M sodium cacodylate buffer) for 4 h, post-fixed in 0.15 M cacodylate buffer and 1% osmium tetroxide for 60 min, and stained in 2% uranyl acetate for 60 min. The samples were treated with pure ethanol for dehydration, resin-embedded, and sectioned to 50–60-nm thickness using the Leica EM UC7 Ultramicrotome (Leica Microsystems, Wetzlar, Germany). Subsequently, 27 grids were viewed using an H7100 Transmission Electron Microscope (Hitachi High-Technologies, Tokyo, Japan), equipped with an XR81 (8 megapixels) charge-coupled device camera (AMT Imaging Systems, Woburn, MA, USA).

## Nanoparticle tracking analysis (NTA)

NanoSight LM10 supplied with fast video capture and NTA software version 2.3 (both Malvern Instruments, Malvern, UK) were used for particle detection, counting, and sizing. The experiment was performed following the manufacturer's instructions.

## Monocyte NF-κB reporter assay

A THP-1 Blue NF-κB reporter monocyte cell suspension of an approximate concentration of $5 \times 10^5$ cells/mL was incubated with 20 μL (approximately $1.0 \times 10^8$ exosomes) of each sample at 37˚C for 18 h. After incubation, 20 μL of sample from each well was extracted and mixed with 180 μL QUANTI-Blue (InvivoGen). The plate was incubated at 37˚C for 4 h, and secreted embryonic alkaline phosphatase levels were quantified using a spectrophotometric microplate reader (VersaMax Microplate Reader, Tokyo, Japan) at 620 nm and SoftMax Pro 5.3 software (Molecular Devices, Silicon Valley, CA, USA).

## Shotgun protein analysis

For protein analysis, a combination of Nano-HPLC Chromatography System and timsTOF Pro (Bruker Daltonics Billerica, MA, USA) was used. To ensure reproducibility, two identical experiments were conducted using separate samples (H1 and H2; samples under hypoxic conditions; N1 and N2, samples under normoxic conditions). The mass spectrometer was operated in the parallel accumulation serial fragmentation mode. Protein identification was conducted using the Mascot software (Matrix Science, Chicago, IL, USA; version 1.7) [19].

## RNA/miRNA isolation and cDNA synthesis

Total RNA was isolated from IEC-derived exosomes using the miRNeasy Mini Kit (Qiagen, Hilden, Germany) following the manufacturer's instructions. The concentration of the isolated RNA was determined by measuring its absorbance at 260 nm using a Nanodrop 1000 spectrophotometer (Thermo Fisher Scientific, Waltham, MA, USA) and a Qubit fluorometer (Thermo Fisher Scientific, Waltham, MA, USA). The extracted RNA was then reverse-transcribed to cDNA using the miScript II RT Kit (Qiagen).

## miRNA PCR array

The gene expression profile of 84 miRNAs was examined using a qRT-PCR assay with the miScript miRNA PCR Array (Human miFinder in 96-well, Qiagen, Germany; 331221-MIHS-001ZD-2) according to the manufacturer's instructions. The relative expression of each miRNA was calculated using the $2^{-\Delta\Delta Ct}$ method. Six housekeeping genes (snoRNAs and the RNU6B snRNA) were used for normalization (S1 Methods).

## HPLC/ESI–MS/MS-based lipidomic analysis

Lipids in the exosomes were extracted using the Bligh and Dyer technique [20]. HPLC/ESI (electrospray ionization)–MS/MS lipidomic analyses were performed using a combination of the QTRAP5500 triple quadrupole-linear ion trap mass spectrometer (AB SCIEX, Foster, CA, USA) and Dionex Ultimate 3000 UHPLC system (Thermo Fisher Scientific). LC separations were achieved using a Develosil C30-UG-3 column (150 × 1.0 mm I.D., 3 μm pore-diameter; Nomura Chemical, Tokyo, Japan). The analyses were performed in the negative ion mode, using multiple reaction monitoring approaches to determine each PC and LPC species. The Analyst Software (AB SCIEX) was used to integrate the LC chromatogram peaks.

## Statistical analysis

Student's *t*-test was performed to assess the differences between samples with and without ischemic stimulation. Volcano plots [21] were also generated for miRNA and protein analysis results. All statistical analyses were performed using R (version 3.6.0; R Foundation for Statistical Computing, Vienna, Austria).

### Bioinformatics analysis

Proteins were analyzed using the Gene Ontology (GO) term enrichment analysis with g:Profiler (https://biit.cs.ut.ee/gprofiler/gost). Protein–protein interaction network functional enrichment analysis was performed using the STRING v11.3 database (https://string-db.org/).

## Results

### Identification and quantification of the collected exosomes

The isolation of exosomes was confirmed using immunoblotting with the Exo-Check Antibody Array, western blotting, TEM, and NTA. Biochemical analysis revealed the presence of exosomes in IECs based on positive staining for ICAM, ALIX, CD81, ANXA5, TSG101, which are common exosomal protein markers (Fig 1A) [22]. The isolated particles exhibited the typical cup-shaped morphology, and the size profiles of exosomes were confirmed using TEM (Fig 1B). Exosomal

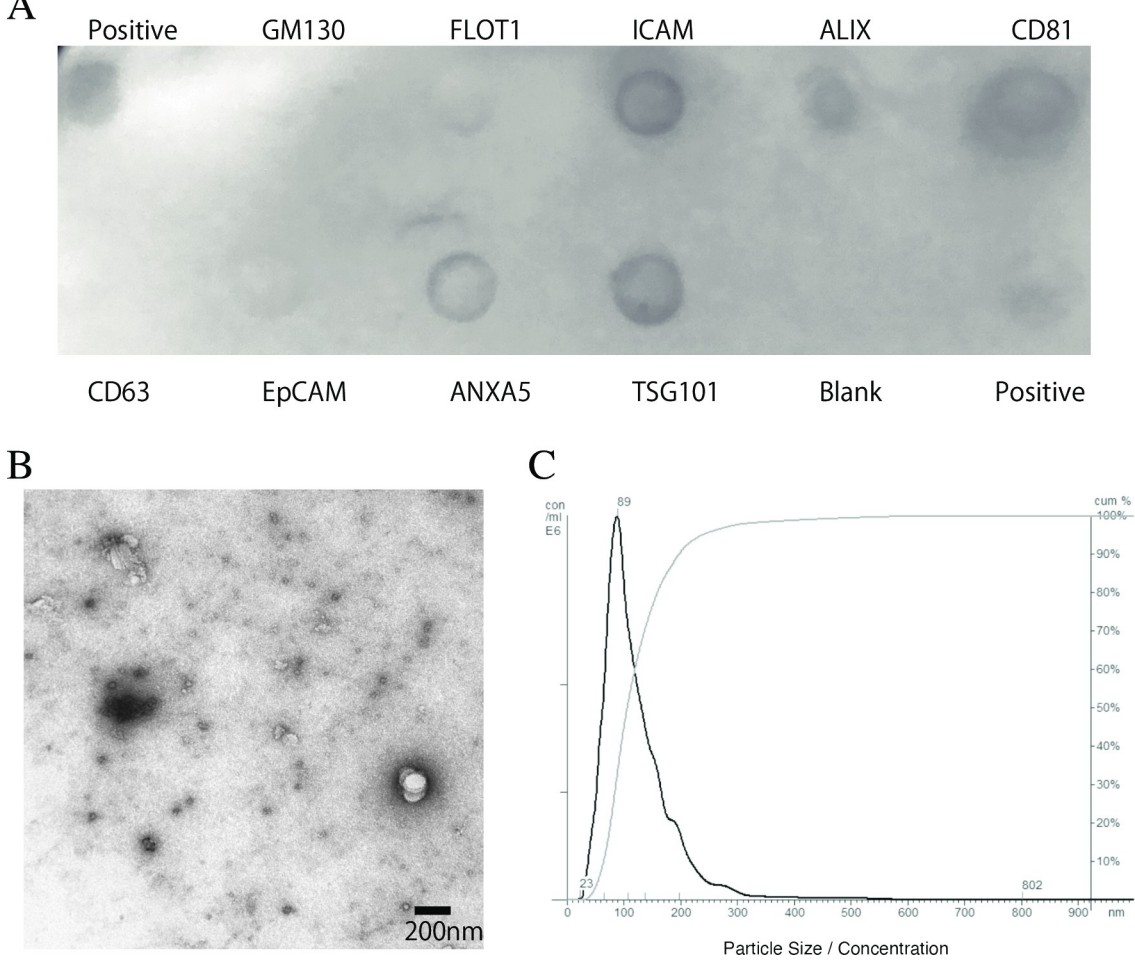

**Fig 1. Identification and morphological characterization of exosomes.** Exosome isolation was confirmed via immunoblotting using the Exo-Check Antibody Array, a transmission electron microscope, and nanoparticle tracking analysis. (**A**) The exosome compartments of the samples were placed in the pre-printed spots of eight antibodies against known exosome markers (FLOT1, ICAM, ALIX, CD81, CD63, EpCAM, TSG101, and ANXA5), in the positive control, and in the blank (negative control) to examine the presence of exosome-specific proteins. (**B**) Representative images from transmission electron microscopy showing typical exosome cup-shaped morphology. (**C**) The size distribution (black line) and cumulative distribution (gray line) of the exosomes were measured using the nanoparticle tracking analysis.

size was determined to be 89 ± 69 nm (mean ± standard deviation), which is consistent with the expected size of exosomes. The size distribution of the exosomes is shown in Fig 1C.

## Changes in the biological activity of exosomes under hypoxic stimulation

As shown in Fig 2, the production of NF-κB increased in the hypoxia group compared with that in the untreated and normoxia groups, indicating that hypoxic stimulation induced the production of exosomes, which activate monocytes.

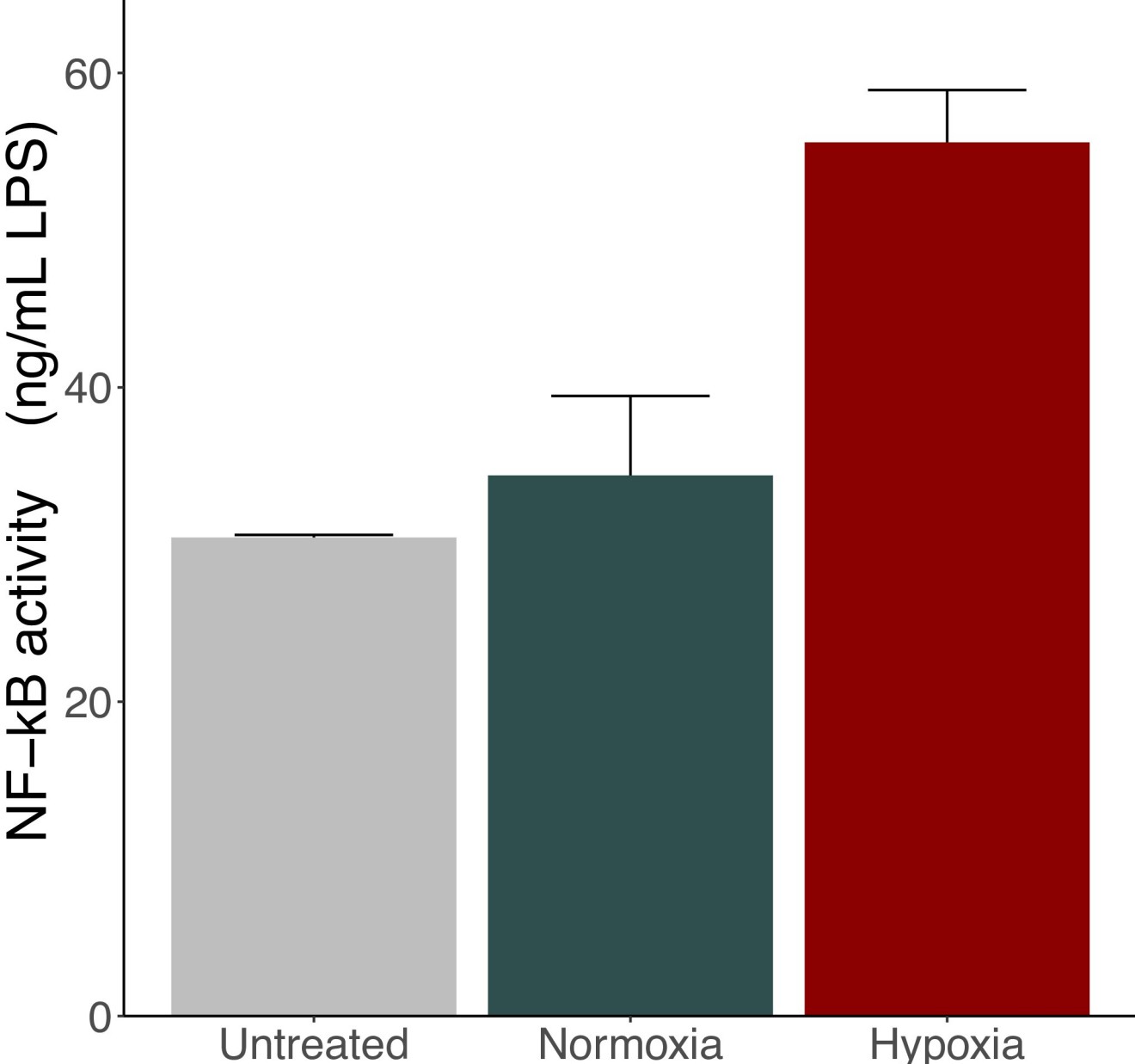

**Fig 2. Effect of intestinal epithelial cell-derived exosomes on NF-κB reporter activation in THP-1 monocytic cells.** The supernatants of the normoxia and hypoxia groups were harvested, and the exosome compartment was extracted. THP-1 Blue NF-κB cells were identically stimulated by the obtained exosomes. The black bar indicates the exosomes obtained from the normoxia group, the red bar indicates the exosomes obtained from the hypoxia group, and the gray bar indicates exosomes obtained from the negative control using 10% (v/v) sterile PBS. Results are expressed as mean ± standard deviation of three experiments. NF-κB, nuclear factor-kappa B.

## Proteomic changes in exosomes derived under hypoxic stimulation

The number of detected proteins in the N1, N2, H1, and H2 samples was 3519, 3407, 3290, and 3127, respectively. Fig 3A presents the changes in protein expression between groups (H1/H2 and N1/N2). Samples in the same group showed almost identical expression patterns. Different patterns were observed between groups, indicating high reproducibility and high experimental precision. Among these identified proteins, 110 showed a >2-fold increase, and 75 showed a <2-fold decrease compared with those in the normoxia group (Fig 3B). Proteins specifically associated with inflammation are labeled in Fig 3B. Their annotations are listed in S1 Table. The proteins whose concentrations largely changed (i.e., >2-fold increase/decrease with $p < 0.01$) are shown in S2 Fig. Among them, proteins associated with inflammation are presented in Fig 4. The protein network analysis showed increased levels of the proteins involved in ubiquitination and subsequent protein degradation (DDX3X, PSMD3, PSMC2, CAND1, and UBR4), and metabolism (C3, ACLY, AGL, PYGL, and NAMPT). Moreover, it showed that they interacted with each other (Fig 3C).

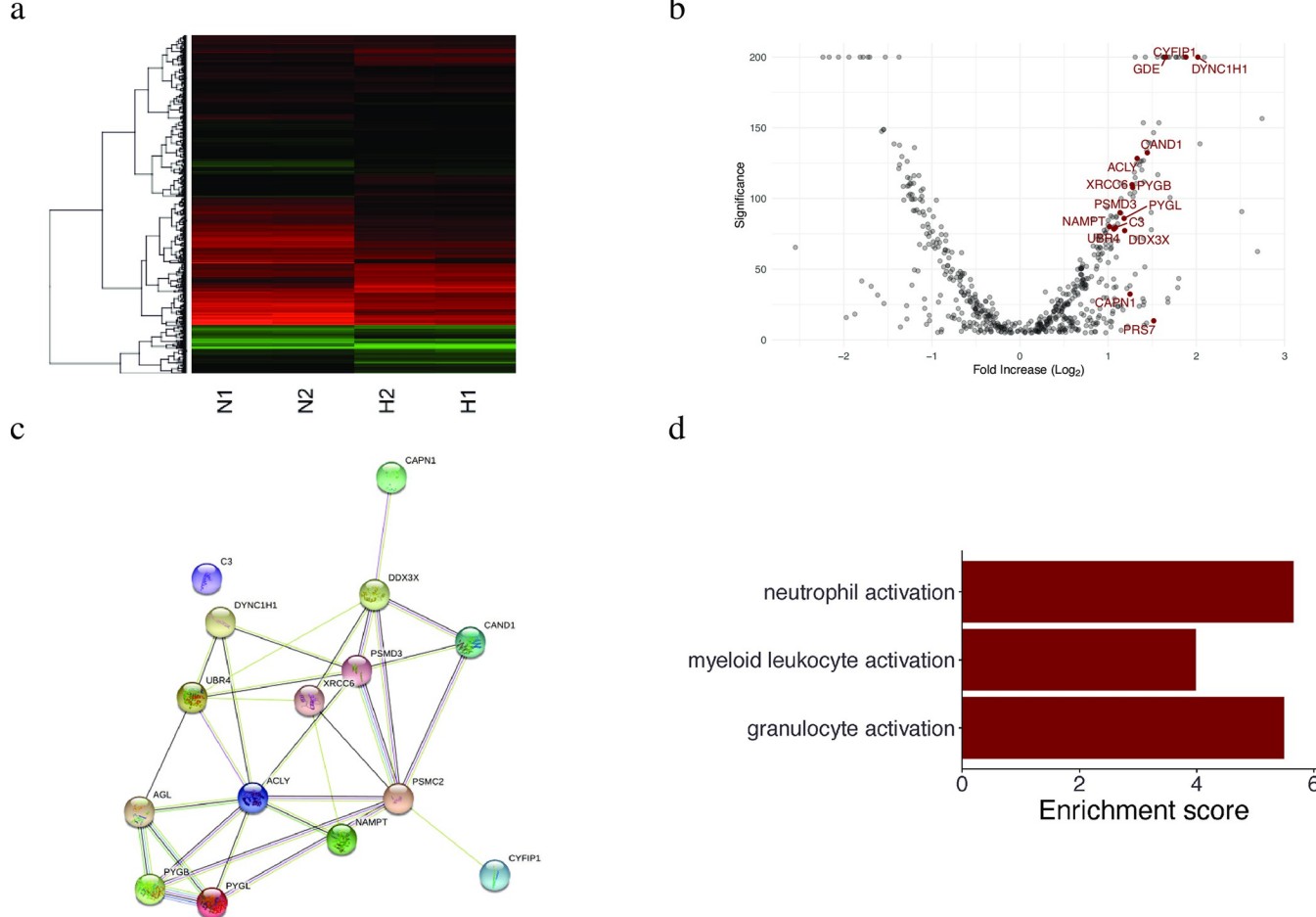

**Fig 3. Proteomic analysis and GO term enrichment analysis of the isolated exosomes.** The expression of proteins in the exosomes was quantified and compared between the normoxia and hypoxia groups using high-performance liquid chromatography-mass spectrometry. The proteins were analyzed using GO term enrichment analysis. (**A**) Heatmap showing protein expression in each group (N1, N2: normoxia; H1, H2: hypoxia). Red indicates high protein expression, whereas green indicates low expression. (**B**) Volcano plot showing the proteins analyzed using mass spectrometry. Red labels indicate proteins involved in inflammation. The description of each protein is given in S1 Table. (**C**) Results of protein network analysis conducted using the STRING v11.3 database. This figure was obtained using STRING v11.3 web-based software (https://string-db.org/). (**D**) Enrichment scores of GO analysis: biological processes. GO, Gene ontology.

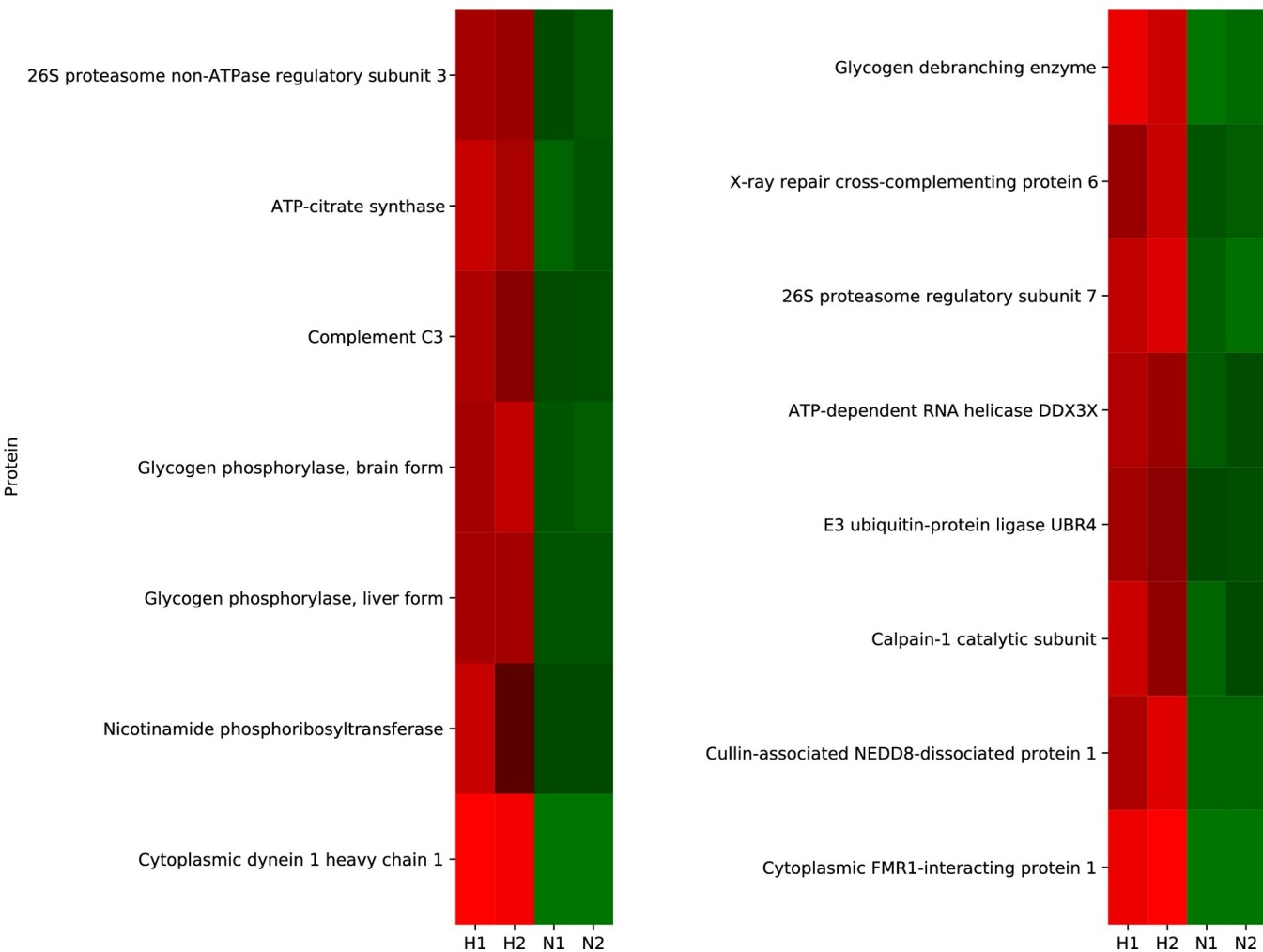

**Fig 4. Heatmap showing the changes in inflammation-related protein expression after ischemic stimulation.** The protein contents of exosomes were extracted and quantified using high-performance liquid chromatography-mass spectrometry. Inflammation-related proteins that showed >2-fold increase/decrease and $p < 0.01$ (based on Student's $t$-test) between the normoxia and hypoxia groups. The changes in the expression of all proteins identified in the experiment are provided in S2 Fig. Values increase progressively from blue to red on the color scale. N1 and N2, normoxia samples; H1 and H2, hypoxia samples.

As shown in Fig 3D, the results of the GO term enrichment analysis showed that these proteins were related to the activation (any process that initiates an immune response) of several components in white blood cells (granulocyte activation, $p = 4.15 \times 10^{-3}$; neutrophil activation, $p = 3.52 \times 10^{-3}$; myeloid leukocyte activation, $p = 1.86 \times 10^{-2}$).

## Changes in miRNA expression in exosomes derived under hypoxic stimulation

Fig 5A presents a volcano plot of the miRNAs obtained from qPCR analysis. Although the absolute change between the normoxia and hypoxia groups was small, a few proteins showed notably low $p$-values. The most prominent miRNAs that showed low $p$-values were hsa-miR-21-5p ($26.75 \pm 0.73$ normoxia vs. $33.98 \pm 0.82$ hypoxia, $p = 6.2 \times 10^{-5}$), hsa-miR-23a-3p ($29.61 \pm 0.64$ normoxia vs. $33.77 \pm 0.85$ hypoxia, $p = 6.2 \times 10^{-3}$), hsa-miR-124-3p ($34.31 \pm 0.29$ normoxia vs. $33.10 \pm 0.45$ hypoxia, $p = 3.5 \times 10^{-3}$), and hsa-miR-30d-5p ($32.81 \pm 0.10$

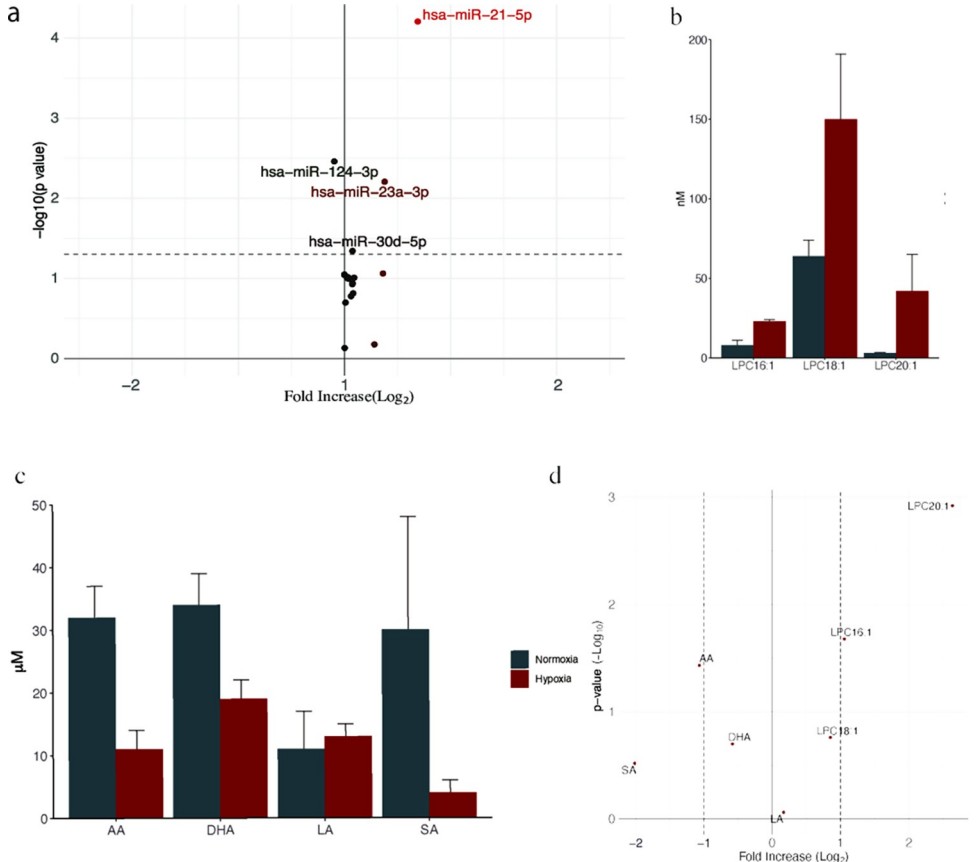

**Fig 5. Changes in the expression of miRNAs and lipids after hypoxic stimulation.** The expression levels of miRNAs extracted from exosomes were measured using a qPCR array and calculated using the $2^{-\Delta\Delta Ct}$ method (three replicates). The lipid compartment in the exosome was extracted using Bligh and Dyer's method. Analyses were performed using high-performance liquid chromatography-mass spectrometry. **(A)** Volcano plot showing miRNA expression plotted using the miScript miRNA PCR Array Data Analysis software. **(B, C)** Changes in the levels of unsaturated lysophosphatidylcholine (**B**) and free fatty acids (**C**) between the normoxia group (black) and hypoxia group (gray). The results are expressed as mean ± standard deviation of three experiments. **(D)** Volcano plot representing unsaturated lysophosphatidylcholine and free fatty acids. LPC, lysophosphatidylcholine; FFA, free fatty acid; AA, arachidonic acid; DHA, docosahexaenoic acid; LA, linoleic acid; SA, stearic acid; miRNA, microRNA.

normoxia vs. 33.70 ± 0.03 hypoxia, $p = 2.8 \times 10^{-2}$; all results are presented as mean ± standard deviation, and $p$-values are based on Student's $t$-test).

## Changes in the lipid profile in exosomes following hypoxic stimulation

The changes in the lipid profile following ischemic stimulation are shown in Fig 5B–5D. The amount of unsaturated LPCs and free fatty acids tended to increase and decrease, respectively, following ischemic stimulation. Fig 5D is a volcano plot of the results representing the increase in the levels of LPC20:1 and LPC16:1 and the decrease in the levels of arachidonic acid.

## Discussion

In this study, using protein, lipid, and miRNA analyses under the same conditions, we gained comprehensive understanding of the changes in exosomal cargo upon ischemic stimulation. The increase in NF-κB activity confirmed that biological activity is induced by hypoxic stimulation. Our results demonstrate that (1) the expression of lipid metabolism-related proteins

was upregulated, (2) the levels of miRNAs were slightly altered, and (3) the concentration of LPCs containing unsaturated fatty acids increased following hypoxia.

The results of our protein analysis were largely different from those of a previous study [13]. In the previous study, the expression of 35 proteins was found to change following ischemia. However, using $p < 0.05$ as the criterion similar to the previous study, our study found the expression of 1632 proteins to be altered [13]. As too many proteins in the present study met the $p < 0.05$ criterion, the 185 proteins that met the $p < 0.01$ and >2-fold increase/decrease criteria were listed for comparison with the proteins identified in the previous study. Of these, only one protein, prosaposin, was common in both studies. This may be due to the high sensitivity of our mass spectrometry analysis. Although previous studies with bioinformatic analyses have identified proteins related to cell survival and metabolism, in this study, we could identify trace levels of proteins related to inflammation and thoroughly examine small protein changes related to inflammation. The protein network analysis showed that the levels of proteins involved in metabolism were increased and that these proteins interacted with each other. ACLY catalyzes acetyl-CoA synthesis from citrate leading to lipid synthesis. NAMPT is the rate-determining enzyme in the mammalian $NAD^+$ biosynthesis salvage pathway that is also essential for maintaining *de novo* lipogenesis [23]. PYGB, AGL, and PYGL degrade glycogen, activate the pentose phosphate pathway, and thereby mediate the production of NADPH, which can also aid *de novo* lipid synthesis [24]. C3 is known to trigger inflammation and can degrade triglycerides [25]. The changes in the miRNA levels were small, which made it difficult to conclude their involvement in inflammation. However, the expression of several miRNAs showed low $p$-values; therefore, their possible influence on the pathology cannot be ruled out. We found marginal upregulation of hsa-miR-21-5p expression. Previous studies have shown that this miRNA is one of the lipid metabolism-related miRNAs, and its upregulation correlates with hyperglycemia, hyperlipidemia, and inflammation markers, such as CRP and IL-1β [26]. hsa-miR-21-5p can also regulate lipid metabolism and mitochondrial metabolism [27]. NF-κB increases the expression of hsa-miR-21-5p, which induces the proliferation of smooth muscle cells in the human pulmonary artery [28, 29]. hsa-miR-21-5p can also induce monocyte differentiation in dendritic cells [30]. These results suggest that hsa-miR-21-5p is a potential therapeutic target for regulating inflammation after ischemic recanalization. In addition, a small increase in miR-23a expression and a decrease in the expression of miR-124-3p were observed. The expression of miR-23a has been reported to decrease in response to NF-κB activation and to enable negative feedback [31], suggesting that an increase in its expression may disrupt this self-regulatory mechanism. Similarly, miR-124-3p exerts anti-inflammatory effects by inducing and maintaining the M2 macrophage phenotype [32, 33]. Therefore, reducing this miRNA level may lead to the dominance of the M1 phenotype in macrophages, causing inflammatory outcomes [33, 34]. Due to the nature of this study, causality cannot be determined but our results suggest that miRNAs and proteins may cooperatively and adversely affect lipid metabolism, leading to induction of the subsequent inflammation. In fact, despite the decrease in free fatty acid levels (arachidonic, docosahexaenoic, and stearic acid), an increase in levels of unsaturated fatty acid containing LPCs (LPC16:1, LPC18:1, LPC20:1) was seen. Increased levels of unsaturated LPCs have been demonstrated to trigger systemic inflammatory response via the NF-κB pathway [12]. As mentioned above, NF-κB increases the expression of hsa-miR-21-5p, which can induce monocyte differentiation in dendritic cells [30–32]. In summary, hypoxic stimuli may be inducing abnormal lipid metabolism by proteins and miRNAs, which triggers NF-kB-mediated inflammation.

There are several limitations to this study. First, this study was conducted *in vitro*, and an experimental system was adopted to simplify the cause of intestinal ischemia. Thus, an *in vivo* study is required to validate our results for clinical translation. Second, although we used

hypoxia stimuli in IECs to mimic the *in vivo* intestinal ischemia-reperfusion injury, this simplified protocol needs to be validated in more detail. Furthermore, as these cells are derived from human adenocarcinoma, their response to stimuli may vary from that of IECs in a physiological state. Third, exosomal RNA concentration measurements were performed using Nanodrop and Qubit, which are less reliable than a bioanalyzer.

## Conclusion

We performed a comprehensive analysis of IEC exosomal contents in hypoxic conditions, which shed light on the pathogenesis of systemic inflammation after intestinal injury. The results of this study have partially clarified the mechanism of multiple organ failure following ischemic recanalization, thereby providing a target for treatment.

## Supporting information

**S1 Table. Annotation of proteins related to inflammation in Gene Ontology terms.** The annotation was obtained from the UniProt database (https://www.uniprot.org/).
(DOCX)

**S1 Fig. Schematic of the experiment.**
(PDF)

**S2 Fig. Heat map showing the changes in protein expression after ischemic stimulation.**
(PDF)

**S1 Methods. List of primers used in the miRNA PCR array.**
(DOCX)

## Acknowledgments

We would like to thank Editage (www.editage.com) for English language editing and Dr. Kana Ariga for creating a schema.

## Author Contributions

**Conceptualization:** Atsushi Senda, Mitsuaki Kojima, Tetsuyuki Kobayashi, Junichi Aiboshi, Yasuhiro Otomo.

**Data curation:** Arisa Watanabe, Tetsuyuki Kobayashi.

**Formal analysis:** Atsushi Senda, Mitsuaki Kojima, Arisa Watanabe.

**Funding acquisition:** Atsushi Senda, Mitsuaki Kojima.

**Investigation:** Atsushi Senda, Arisa Watanabe, Tetsuyuki Kobayashi, Junichi Aiboshi.

**Methodology:** Atsushi Senda, Arisa Watanabe, Tetsuyuki Kobayashi, Junichi Aiboshi.

**Project administration:** Atsushi Senda, Koji Morishita, Junichi Aiboshi.

**Resources:** Atsushi Senda.

**Supervision:** Tetsuyuki Kobayashi.

**Validation:** Atsushi Senda, Mitsuaki Kojima, Tetsuyuki Kobayashi.

**Visualization:** Atsushi Senda, Yasuhiro Otomo.

**Writing – original draft:** Atsushi Senda, Tetsuyuki Kobayashi, Junichi Aiboshi, Yasuhiro Otomo.

**Writing – review & editing:** Atsushi Senda, Mitsuaki Kojima, Tetsuyuki Kobayashi, Koji Morishita, Junichi Aiboshi, Yasuhiro Otomo.

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
