## [Decision Letter · Decision Letter 0]

24 Nov 2022

PONE-D-22-23365Multi-omics profiles of exosomes derived from intestinal epithelial cells under ischemia–reperfusion injuryPLOS ONE

Dear Dr. Kojima,

Thank you for submitting your manuscript to PLOS ONE. After review we feel that it has merit but requires addressed various concerns of the reviewers prior to meeting PLOS ONE’s publication criteria. Therefore, we invite you to submit a revised version of the manuscript that addresses the points raised during the review process. Please address each point made by the reviewers specifically as noted below.  

We look forward to receiving your revised manuscript.

Kind regards,

Jon M. Jacobs, Ph.D.

Academic Editor

PLOS ONE

Journal Requirements:

“NO authors have competing interests”

4. PLOS ONE now requires that authors provide the original uncropped and unadjusted images underlying all blot or gel results reported in a submission’s figures or Supporting Information files. This policy and the journal’s other requirements for blot/gel reporting and figure preparation are described in detail at https://journals.plos.org/plosone/s/figures#loc-blot-and-gel-reporting-requirements and https://journals.plos.org/plosone/s/figures#loc-preparing-figures-from-image-files. When you submit your revised manuscript, please ensure that your figures adhere fully to these guidelines and provide the original underlying images for all blot or gel data reported in your submission. See the following link for instructions on providing the original image data: https://journals.plos.org/plosone/s/figures#loc-original-images-for-blots-and-gels. In your cover letter, please note whether your blot/gel image data are in Supporting Information or posted at a public data repository, provide the repository URL if relevant, and provide specific details as to which raw blot/gel images, if any, are not available. Email us at plosone@plos.org if you have any questions

Reviewers' comments:

Reviewer's Responses to Questions

**Comments to the Author**

1. Is the manuscript technically sound, and do the data support the conclusions?

Reviewer #1: Partly

Reviewer #2: No

2. Has the statistical analysis been performed appropriately and rigorously? 

Reviewer #1: I Don't Know

Reviewer #2: Yes

3. Have the authors made all data underlying the findings in their manuscript fully available?

Reviewer #1: Yes

Reviewer #2: Yes

4. Is the manuscript presented in an intelligible fashion and written in standard English?

Reviewer #1: Yes

Reviewer #2: Yes

5. Review Comments to the Author

Reviewer #1: Multi-omics profiles of exosomes derived from intestinal epithelial cells under ischemia–reperfusion injury

Title

A major focus of the recently updated guidelines of the International Society for Extracellular Vesicles (ISEV) on minimal information for studies of extracellular vesicles (MISEV) is the need for appropriate nomenclature in reporting extracellular vesicle (EV) research. So I advise the authors to use a more precise term

Abstract

Many unimportant technical details are mentioned in the abstract. I advise the authors get rid of them.

Introduction

The introduction did not sufficiently shed light on the studies related to hypoxia and its effect on the components of exosomes.

Methods:

The mentioned method is not well explained. I believe that hypoxia tests should be conducted in a standard atmosphere such as the hypoxia chamber.

The authors mentioned that the cell culture was free of serum, what is the reason?

ExoQuick-TC Exosome Precipitation Solution Kit is not an accurate method for isolating exosomes. So, the authors should use a more precise phrase. For example, the concentration of extracellular vesicles.

The authors did not clarify the amount of media used to obtain a sufficient amount of exosomes.

In the Exo-Check Antibody Array, its method is not well explained. As well as the positive sample, which appears to have not worked, shows a clear result.

Nanodrop are not a valid method for measuring exosomal RNA concentration and quality. It often shows inaccurate results and residual salts from the precipitate. The best way is by bioanalyzer

How the authors ensured that their exosome content was not contaminated with some of the components present in the media. Especially since the isolation was done here by precipitation.

Results:

Isolation and characterization of exosomes is a major and important experiment. So it is supposed to be in the main body of the manuscript and not in the appendices.

Discussion

The discussion sheds light on the possible role and impact of exosomes after hypoxia. This discussion was not deep enough.

Reviewer #2: Atsushi Senda et al. investigated whether the isolated intestinal epithelial cell-derived exosomes ameliorated intestinal epithelia cells hypoxia reoxygenation injury. I have some concerns.

The data without in vivo experiment cannot support any valuable conclusion for clinical translation.

The study is too simple and too observational. It needs more mechanism investigation.

6. PLOS authors have the option to publish the peer review history of their article (what does this mean?). If published, this will include your full peer review and any attached files.

Reviewer #1: **Yes: **Faisal A A

Reviewer #2: No

---

## [Author Response · Author response to Decision Letter 0]

17 Feb 2023

RESPONSE TO REVIEWERS’ COMMENTS

We would like to thank the reviewers for their insightful comments and suggestions. Please see our point-by-point responses to reviewers’ comments below. Our responses are all labelled as “Response.”

Reviewer #1: Multi-omics profiles of exosomes derived from intestinal epithelial cells under ischemia–reperfusion injury

Title

A major focus of the recently updated guidelines of the International Society for Extracellular Vesicles (ISEV) on minimal information for studies of extracellular vesicles (MISEV) is the need for appropriate nomenclature in reporting extracellular vesicle (EV) research. So I advise the authors to use a more precise term

Response: We thank the reviewer for their suggestion. We have changed the title to “Profiles of lipid, protein and microRNA expression in exosomes derived from intestinal epithelial cells after ischemia-reperfusion injury in a cellular hypoxia model.”

Abstract

Many unimportant technical details are mentioned in the abstract. I advise the authors get rid of them.

Response: We thank the reviewer for their comment. We agree and have reduced the description of technical details in the abstract as suggested.

Introduction

The introduction did not sufficiently shed light on the studies related to hypoxia and its effect on the components of exosomes.

Response: We thank the reviewer for their constructive comments. We have added the following sentence to the third paragraph of the Introduction in line with their suggestion to show the findings between hypoxia and its effects on exosome components:

Corresponding to the ischemic state, cultured IECs were first placed under hypoxic conditions. Subsequently, they were returned to an oxygenated environment corresponding to the reperfusion state. Accordingly, this hypoxia reoxygenation model has been used to simulate ischemia-reperfusion injury in an in vitro setting in previous studies [14,15]. 

14. Zw D, H L, Ff S, Xz F, Y Z, P L. Cystic fibrosis transmembrane conductance regulator prevents ischemia/reperfusion induced intestinal apoptosis via inhibiting PI3K/AKT/NF-κB pathway. World journal of gastroenterology. 2022;28. doi:10.3748/wjg.v28.i9.918

15. Kip AM, Soons Z, Mohren R, Duivenvoorden AAM, Röth AAJ, Cillero-Pastor B, et al. Proteomics analysis of human intestinal organoids during hypoxia and reoxygenation as a model to study ischemia-reperfusion injury. Cell Death Dis. 2021;12: 95. doi:10.1038/s41419-020-03379-9

Methods:

The mentioned method is not well explained. I believe that hypoxia tests should be conducted in a standard atmosphere such as the hypoxia chamber.

Response: To generate an environment analogous to that of the reperfusion period in hemorrhagic shock, cells in the hypoxia and normoxia groups were subsequently incubated in the normoxic environment for 24 h. All tests, including biological activity tests, proteome analyses, lipid analyses, and miRNA quantification, were performed in a normoxic environment. We have made appropriate changes to the revised manuscript to clarify this point.

The authors mentioned that the cell culture was free of serum, what is the reason?

Response: We thank the reviewer for their question. We used serum-free cell culture to avoid the potential contamination of the exosomes that could derive from the serum. We have changed our description in the revised manuscript to ensure that readers understand this point.

ExoQuick-TC Exosome Precipitation Solution Kit is not an accurate method for isolating exosomes. So, the authors should use a more precise phrase. For example, the concentration of extracellular vesicles.

The authors did not clarify the amount of media used to obtain a sufficient amount of exosomes.

Response: We would like to thank the reviewer for raising this point. We used 80 mL of media to obtain sufficient amount of exosomes. We have added this clarification to our revised manuscript. 

In the Exo-Check Antibody Array, its method is not well explained. As well as the positive sample, which appears to have not worked, shows a clear result.

Response: We have included a more detailed explanation in the section kindly pointed out. Our results were positive for ICAM, ALIX, CD81, ANXA5, TSG101, and positive control; intermediate for FLOT1 and EpCAM; and negative for GM130, CD63, and Blank. The difference in concentration between the two positive controls may be caused by a concentration irregularity in the sample or in the blocking agent.

As for CD63 expression, the amount and type of surface antigens depends on the cell type from which the exosomes are extracted. In addition, CD63 may not be detected using the kit due to the reactivity between the antibody and cell type. The figure below (manufacturer's manual) also shows that the surface markers on exosomes in human serum and HEK cells are very different. We believe that the relatively low distribution of CD63 on the cell surface of exosomes extracted from the Caco-2 cells used in this study could not be detected, unlike other markers such as CD81. Since this is a simple kit test, if most of the exosome markers are detected, it is considered evidence for the presence of exosomes.

Figures from Exo-Check Antibody Array User-manual P.5

http://www.systembio.com/wp/wp-content/uploads/Exo-Check_User-Manual.pdf

Nanodrop are not a valid method for measuring exosomal RNA concentration and quality. It often shows inaccurate results and residual salts from the precipitate. The best way is by bioanalyzer

Response: We appreciate the reviewer’s highly appropriate remarks. To avoid the complexity of the description, we originally referred to this as Nanodrop, but in the actual measurement, we used both Qubit and Nanodrop. As pointed out, there were samples that showed unexpected values with Nanodrop; thus, we did not perform further measurements on them. In the revised manuscript, we have amended the Methods section regarding this point and added text that we did not use a bioanalyzer as a study limitation.

How the authors ensured that their exosome content was not contaminated with some of the components present in the media. Especially since the isolation was done here by precipitation.

Response: We are deeply concerned about the points raised by the reviewer. We used serum-free media to avoid contamination. However, as the reviewer pointed out, the existence of contamination cannot be completely ruled out. Therefore, we decided to present a relative evaluation between the hypoxia group and normoxia group in our experimental system:

In a preliminary experiment, the protein concentration and exosome particle size distribution were compared between exosome samples subjected to centrifugation at 10,000 × g and those subjected to the ExoQuick-TC followed by 0.2-µm filtration. As pointed out by the reviewer, the amount of 30–150-nm particles was lower in the exosomes obtained using ExoQuick followed by filtration than in those obtained with ultracentrifugation (82% in ultracentrifuge versus 78% ExoQuick, please refer to the figure below).

To determine whether the above difference is significant, we compared the protein amount in exosomes obtained by ultracentrifuge with that in the exosomes obtained by ExoQuick. 

Additionally, as shown in the figure below, the correlation coefficients between the protein amounts between the Ultracentrifuge and ExoQuick groups were over 99.5% (99.3% between ultracentrifuge-isolated exosome compartment and ExoQuick-isolated exosome compartment in the hypoxia group, and 99.6% in the normoxia group).Therefore, we concluded that there was no significant difference between the ExoQuick and ultracentrifugation results and chose to perform the experiment using the Centrifugal Filter Unit for better experimental efficiency.

Results:

Isolation and characterization of exosomes is a major and important experiment. So it is supposed to be in the main body of the manuscript and not in the appendices.

Response: We agree with the comment and have revised our manuscript as suggested. Please see Figure 1.

Discussion

The discussion sheds light on the possible role and impact of exosomes after hypoxia. This discussion was not deep enough.

Response: We thank the reviewer for this useful suggestion. Since this study is only an observation of qualitative relationships, we have added some more to the Discussion section, but being mindful not to exceed the extent to which conclusions can be stated.

Reviewer #2: Atsushi Senda et al. investigated whether the isolated intestinal epithelial cell-derived exosomes ameliorated intestinal epithelia cells hypoxia reoxygenation injury. I have some concerns.

The data without in vivo experiment cannot support any valuable conclusion for clinical translation. The study is too simple and too observational. It needs more mechanism investigation.

Response: We appreciate the valuable comments made by the reviewer. To better understand the significance of this study, please allow us to explain the background of our previous studies.

We have focused on in vivo studies using animal models. As described in the background, we focused on in vitro experiments to prove the following assumptions.

1. We first confirmed the presence of exosomes in the lymph fluid of rats in 2017[1].

2. We have shown that exosomes in mesenteric lymph fluid have biological activity after hemorrhagic shock in an animal model and that these exosomes induce lung injury when administered to the naive animal [2].

3. Exosomes extracted from rat lymph fluid in a rat intestinal ischemia-reperfusion model have been shown to induce immune cell inflammation, as in the trauma hemorrhagic shock model [3].

4. The lipid concentration in exosomes was altered after ischemia-reperfusion injury, and some lipids, such as polyunsaturated fatty acids and lysophosphatidylcholines, revealed significant changes in exosomes [3].

The above in vivo studies suggest that inflammatory exosomes extracted from the intestinal tract travel through the lymphatic system to cause inflammation in the lungs and other organs. However, lymphatic fluid contains not only exosomes derived from intestinal cells, but also exosomes secreted from vascular endothelial cells and blood cell components, and not only exosomes derived from intestinal epithelial cells. Therefore, in this study we decided to focus only on intestinal epithelial cells to clarify how their composition is altered by hypoxic stimuli and how they are involved in inflammation.

We fully agree with the reviewer's point. However, we have now presented an in vitro focused study with the aim of selecting targets for future animal studies by analyzing lipids, proteins, and miRNAs in detail.

We appreciate the valuable comments made by the reviewer. To better contextualize the significance of this study, we would like to provide additional information about our previous research. Our previous work has mainly focused on in vivo studies using animal models. As described in the background, we conducted in vitro experiments to confirm the following hypotheses:

1. We confirmed the presence of exosomes in the lymph fluid of rats in 2017 [1].

2. We demonstrated that exosomes in mesenteric lymph fluid have biological activity after hemorrhagic shock in an animal model, and that these exosomes induce lung injury when administered to naive animals [2].

3. Exosomes extracted from rat lymph fluid in a rat intestinal ischemia-reperfusion model induced immune cell inflammation, similar to the trauma hemorrhagic shock model [3].

4. Lipid concentration in exosomes was found to be altered after ischemia-reperfusion injury, with significant changes in polyunsaturated fatty acids and lysophosphatidylcholines [3].

The above in vivo studies suggest that inflammatory exosomes extracted from the intestinal tract may cause inflammation in the lungs and other organs. However, since lymphatic fluid contains exosomes from various sources, including vascular endothelial cells and blood cell components, we decided to focus on intestinal epithelial cell-derived exosomes in this study to investigate how their composition is altered by hypoxic stimuli and their potential role in inflammation.

We agree with the reviewer's point that our study is limited in scope and lacks in vivo experiments. However, we present an in vitro study to provide insights into the mechanisms underlying the protective effects of intestinal epithelial cell-derived exosomes against hypoxia reoxygenation injury, which could serve as a basis for future animal studies. We analyzed the lipids, proteins, and miRNAs in detail to identify potential targets for future studies.

References

1. Kojima M, Gimenes-Junior JA, Langness S, Morishita K, Lavoie-Gagne O, Eliceiri B, et al. Exosomes, not protein or lipids, in mesenteric lymph activate inflammation: Unlocking the mystery of post-shock multiple organ failure. J Trauma Acute Care Surg. 2017;82: 42–50. doi:10.1097/TA.0000000000001296

2. Kojima M, Gimenes-Junior JA, Chan TW, Eliceiri BP, Baird A, Costantini TW, et al. Exosomes in postshock mesenteric lymph are key mediators of acute lung injury triggering the macrophage activation via Toll-like receptor 4. FASEB J. 2018;32: 97–110. doi:10.1096/fj.201700488R

3. Senda A, Morishita K, Kojima M, Doki S, Taylor B, Yagi M, et al. The role of mesenteric lymph exosomal lipid mediators following intestinal ischemia-reperfusion injury on activation of inflammation. J Trauma Acute Care Surg. 2020;89: 1099–1106. doi:10.1097/TA.0000000000002897

---

## [Decision Letter · Decision Letter 1]

14 Mar 2023

Profiles of lipid, protein and microRNA expression in exosomes derived from intestinal epithelial cells after ischemia-reperfusion injury in a cellular hypoxia model

PONE-D-22-23365R1

Dear Dr. Kojima,

We’re pleased to inform you that your manuscript has been judged scientifically suitable for publication and will be formally accepted for publication once it meets all outstanding technical requirements.

Kind regards,

Jon M. Jacobs, Ph.D.

Academic Editor

PLOS ONE

Additional Editor Comments (optional):

Reviewers' comments:

Reviewer's Responses to Questions

**Comments to the Author**

1. If the authors have adequately addressed your comments raised in a previous round of review and you feel that this manuscript is now acceptable for publication, you may indicate that here to bypass the “Comments to the Author” section, enter your conflict of interest statement in the “Confidential to Editor” section, and submit your "Accept" recommendation.

Reviewer #2: All comments have been addressed

2. Is the manuscript technically sound, and do the data support the conclusions?

Reviewer #2: Yes

3. Has the statistical analysis been performed appropriately and rigorously? 

Reviewer #2: Yes

4. Have the authors made all data underlying the findings in their manuscript fully available?

Reviewer #2: Yes

5. Is the manuscript presented in an intelligible fashion and written in standard English?

Reviewer #2: Yes

6. Review Comments to the Author

Reviewer #2: I have no more comment. My concerns have been addressed. I suggest language editing before fully acceptance.

7. PLOS authors have the option to publish the peer review history of their article (what does this mean?). If published, this will include your full peer review and any attached files.

Reviewer #2: No

---

## [Editor Report · Acceptance letter]

20 Mar 2023

PONE-D-22-23365R1 

Profiles of lipid, protein and microRNA expression in exosomes derived from intestinal epithelial cells after ischemia-reperfusion injury in a cellular hypoxia model 

Dear Dr. Kojima:

I'm pleased to inform you that your manuscript has been deemed suitable for publication in PLOS ONE. Congratulations! Your manuscript is now with our production department. 

Kind regards, 

on behalf of

Dr Jon M. Jacobs 

Academic Editor

PLOS ONE